# Development of a pentavalent broadly protective nucleoside-modified mRNA vaccine against influenza B viruses

Norbert Pardi [1] ✉, Juan Manuel Carreño[2,3], George O'Dell[2], Jessica Tan [2,4], Csaba Bajusz[1,5], Hiromi Muramatsu[1], Willemijn Rijnink[2], Shirin Strohmeier[2], Madhumathi Loganathan [2,3], Dominika Bielak[2,3], Molly M. H. Sung [6], Ying K. Tam[6], Florian Krammer [2,7] ✉ & Meagan McMahon [2] ✉

Messenger RNA (mRNA) vaccines represent a new, effective vaccine platform with high capacity for rapid development. Generation of a universal influenza virus vaccine with the potential to elicit long-lasting, broadly cross-reactive immune responses is a necessity for reducing influenza-associated morbidity and mortality. Here we focus on the development of a universal influenza B virus vaccine based on the lipid nanoparticle-encapsulated nucleoside-modified mRNA (mRNA-LNP) platform. We evaluate vaccine candidates based on different target antigens that afford protection against challenge with ancestral and recent influenza B viruses from both antigenic lineages. A pentavalent vaccine combining all tested antigens protects mice from morbidity at a very low dose of 50 ng per antigen after a single vaccination. These findings support the further advancement of nucleoside-modified mRNA-LNPs expressing multiple conserved antigens as universal influenza virus vaccine candidates.

While a large proportion of influenza virus research focuses on influenza A viruses (IAVs), the public health concern caused by influenza B viruses (IBVs) can no longer be overlooked. The larger proportion of human influenza virus infections attributed to IAVs, the pandemic potential of IAVs, and the long-held misconceptions regarding the severity and impact of infections caused by IBVs have contributed to establishing a historical trend towards research focusing on IAVs. Recently, however, many studies have conclusively demonstrated the significant burden of IBV infections as a global health concern[1,2]. Further complicating IBV infections is the separation of IBVs into two distinct lineages (B/Yamagata/16/1988-like and B/Victoria/2/1987-like), with delineation based on the sequences of the hemagglutinin (HA), the immunodominant surface glycoprotein of influenza viruses. In addition, the B/Yamagata/16/1988-like lineage (Y) recently split into

multiple clades and B/Victoria/2/1987-like lineage (V) viruses with amino acid deletions have emerged[3], adding to antigenic diversity.

Current quadrivalent seasonal influenza vaccines (QIVs) include representative strains from both IBV lineages. These vaccines are focused on eliciting an antibody response towards the HA. Despite the availability of QIVs that include both IBV lineages, QIVs continue to constitute a minority of the influenza virus vaccines administered globally as countries consider the cost-effectiveness and fiduciary impact of increasing vaccine valency not covered in trivalent vaccines (where only one IBV lineage is included)[2,4]. In addition, antigenic drift of the HA of circulating viruses can render vaccine-induced antibodies ineffective, significantly undermining vaccine effectiveness. As such, IAV and IBV strains to be included in seasonal vaccines need to be updated annually based on surveillance and predictions. Mismatches

[1]Department of Microbiology, Perelman School of Medicine, University of Pennsylvania, Philadelphia, PA 19104, USA. [2]Department of Microbiology, Icahn School of Medicine at Mount Sinai, New York, NY 10029, USA. [3]Center for Vaccine Research and Pandemic Preparedness (C-VARPP), Icahn School of Medicine at Mount Sinai, New York, NY, USA. [4]Graduate School of Biomedical Sciences, Icahn School of Medicine at Mount Sinai, New York, NY 10029, USA. [5]Biotechnological National Laboratory, Institute of Genetics, Biological Research Centre, Szeged 6726, Hungary. [6]Acuitas Therapeutics, Vancouver, BC V6T 1Z3, Canada. [7]Department of Pathology, Molecular and Cell Based Medicine, Icahn School of Medicine at Mount Sinai, New York, NY 10029, USA. ✉e-mail: pnorbert@pennmedicine.upenn.edu; florian.krammer@mssm.edu; meagankmcmahon@gmail.com

between vaccine strains and circulating strains can still occur and result in decreased vaccine effectiveness, creating an urgent need for new vaccines and treatment options that can provide broader and more durable protection against the ever-evolving influenza viruses.

The development of vaccination regimens that target multiple, conserved epitopes of IBVs has mostly been limited to assessing combinations of HA and neuraminidase (NA)[5–7]. Conserved regions within these IBV antigens can act as targets for the induction of broadly protective humoral responses. The HA has been the object of much attention due to its ability to induce protection via hemagglutination inhibition, virus neutralization, and Fc effector functions[7]. The IBV NA has also raised considerable interest after antibodies to this protein were found to provide protection across all influenza B lineages in mice and broadly reactive influenza virus NA-specific antibodies have been isolated from human donors[5,8,9]. Two highly conserved IBV proteins, the matrix-2 (M2) ion channel protein and nucleoprotein (NP) of the IBVs have been understudied when compared to IAV antigens. Targeting these antigens in IAV vaccination studies has been relatively successful as these antigens can induce broadly protective immune responses through antibody Fc-mediated mechanisms and cellular responses in the context of M2 vaccinations, and cellular responses following NP vaccination[10–12].

Our previous work with nucleoside-modified mRNA-LNP vaccines demonstrated that simultaneous targeting of the A/Brisbane/59/2007 H1N1 HA using a headless HA construct, a membrane-bound A/Michigan/45/2015 NA, A/Michigan/45/2015 NP and A/Michigan/45/2015 M2 with a quadrivalent formulation provided broadly protective immunity in mice[13]. In a follow-up study we showed that certain antigen modifications yielded more potent and less reactogenic mRNA-LNP influenza virus vaccines[14].

Since coronavirus disease 2019 (COVID-19) nucleoside-modified mRNA-LNP vaccines proved to be safe and very effective in humans, we believe that this platform should be further assessed for its potential to generate a potent, broadly protective influenza virus vaccine for humans[15,16]. In the studies presented here, we again harnessed the nucleoside-modified mRNA-LNP technology to effectively deliver a pentavalent influenza B vaccine candidate that targets a combination of antigens (B/Yamagata/16/1988-like lineage HA, B/Victoria/2/1987-like lineage HA, NA, NP, and M2) and provides broad protection in mice after administration of a single, low dose.

## Results

### Selection of IBV vaccine antigens and cell transfection studies

IBVs are classified into two lineages (B/Yamagata/16/1988-like (Y) and B/Victoria/2/1987-like (V)) based on the antigenic properties of the HA glycoprotein. NA, NP, and M2 are fairly conserved in these strains. Therefore, we utilized mRNA-encoded HAs from both lineages (B/Phuket/3073/2013 (Y) and B/Colorado/06/2017 (V)) and NA, NP, and M2 from the B/Colorado/06/2017 strain. Before the mRNA-LNP vaccines were made, protein production from mRNA vaccine antigens was confirmed in cell transfection studies. HEK293T cells were transfected with B/Phuket/3073/2013 (B/Phu) HA, B/Colorado/06/17 (B/Col) HA, B/Col NA, and B/Col NP antigens and expression was assessed via flow cytometry or Western blotting. Binding of CR8033 and CR8059 anti-HA antibodies was observed for B/Phu HA- and B/Col HA-transfected HEK293T cells, respectively, via flow cytometry (Fig. 1a, b). We also observed binding of monoclonal antibody (mAb) 1G05, an anti-NA active site antibody[9], to B/Col NA-transfected cells via flow cytometry (Fig. 1c).

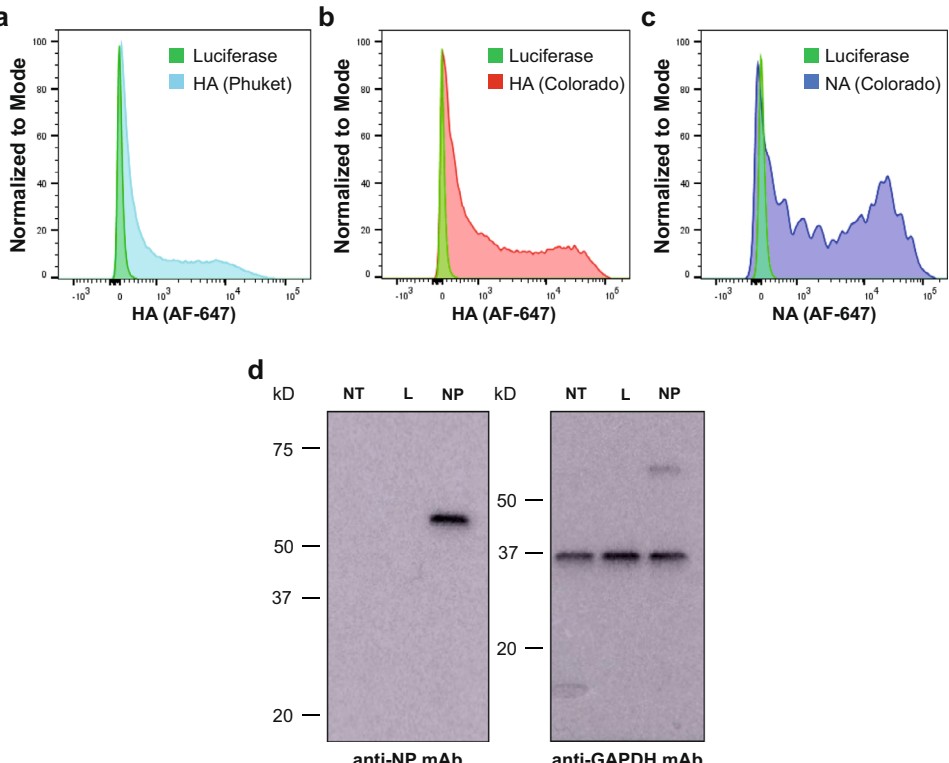

**Fig. 1 | Characterization of antigen-encoding mRNAs by flow cytometry or Western blot.** HEK 293 T cells were transfected with HA-, NA- or luciferase control-encoding mRNAs and protein production from each influenza virus antigen-encoding mRNA was assessed via flow cytometry. Positive binding of the antibodies specific for B/Phu HA (**a**), B/Col HA (**b**), and B/Col NA (**c**) relative to luciferase control. HEK 293 T cells were transfected with B/Col NP- or luciferase-encoding mRNAs and protein production from NP-encoding mRNA was assessed via Western blotting. All experiments were performed once. (**d**) NT non-transfected, L luciferase, GAPDH glyceraldehyde 3-phosphate dehydrogenase. Source data are provided as a source data file.

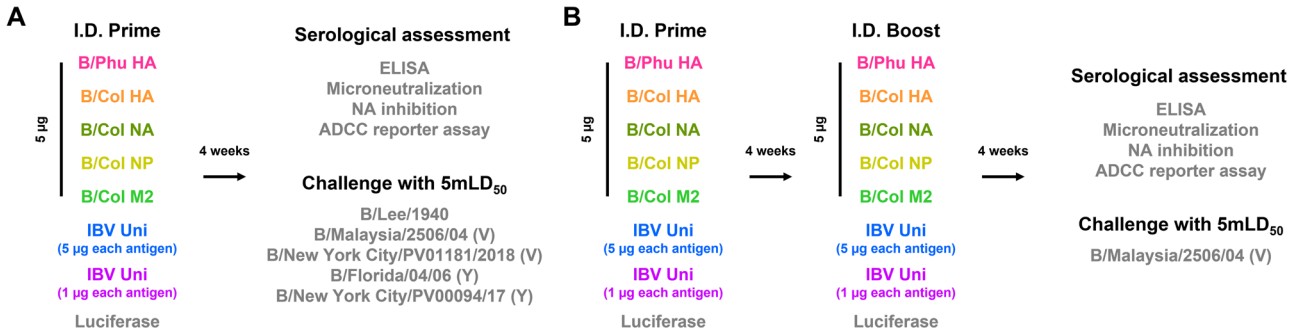

**Fig. 2 | Experimental plan: vaccines, immunizations, serology and challenge studies.** Mice received a single I.D. immunization with 5 µg monovalent mRNA-LNP vaccine or 5 µg (1 µg from each component) or 25 µg (5 µg from each component) of the pentavalent mRNA-LNP vaccine. Animals in the negative control group received 5 µg luciferase mRNA-LNP. Four weeks after vaccination, mice were intranasally (I.N.) infected with influenza virus containing 1x or 5x the mLD$_{50}$. Additionally, mice were bled for serological analysis (ELISA, MNT, NAI, and ADCC reporter assay) at this timepoint (**a**). For prime-boost vaccination studies mice were vaccinated twice as described with a 4-week interval between vaccinations. Four weeks after the boost, mice were I.N. challenged with 5mLD$_{50}$ of B/Malaysia/2506/2004 virus. Additionally, mice were bled for serological analysis (ELISA, MNT, NAI, and ADCC reporter assay) at this time point (**b**).

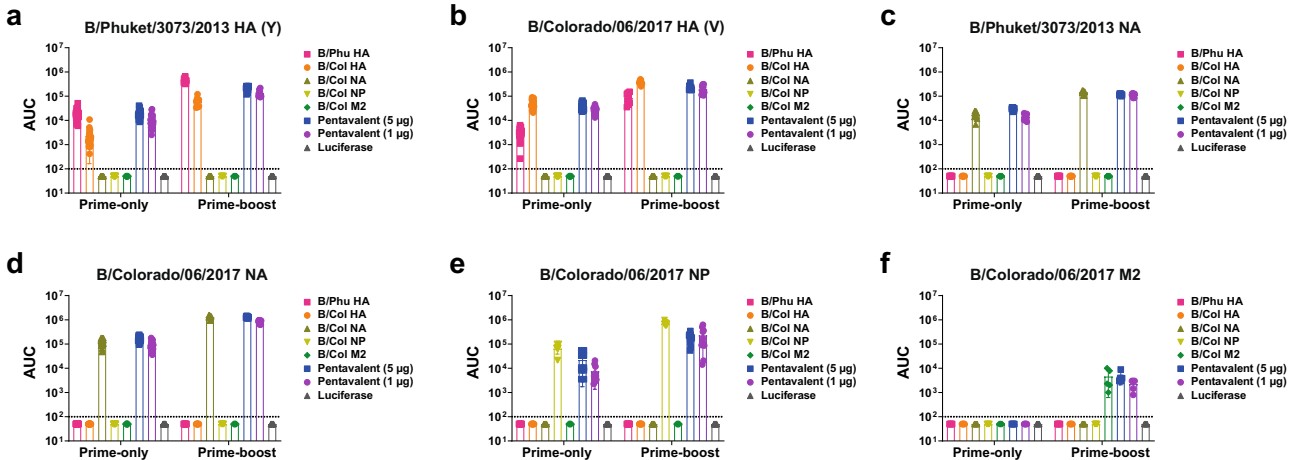

**Fig. 3 | Serum reactivity to different IBV antigens.** Mice were vaccinated with a prime only or prime-boost vaccination regimen (28 days apart) I.D. with 5 µg of monovalent mRNA-LNPs or with 5 µg/antigen or 1 µg/antigen of the pentavalent mRNA-LNP formulation. Control animals received 5 µg of luciferase mRNA-LNP. Sera were collected on day 28 post prime and post boost and binding of antibodies to influenza antigens was measured by ELISA. Binding of sera against B/Phuket/307/2013 HA ($n$ = 25 prime-only samples, $n$ = 10 prime-boost samples). (**a**) B/Colorado/ 06/2017 HA ($n$ = 25 prime-only samples, $n$ = 10 prime-boost samples). (**b**) B/Phuket/ 307/2013 NA ($n$ = 10) (**c**) B/Colorado/06/2017 NA ($n$ = 25 prime-only samples, $n$ = 10 prime-boost samples) (**d**) B/Colorado/06/2017 NP ($n$ = 10) (**e**) and B/Colorado/06/ 2017 M2 ($n$ = 5) (**f**). Each symbol represents one animal. AUCs with a cutoff value of the average background plus three SDs are shown. Bars represent the mean of each group. The dotted lines indicate the limit of detection. Source data are provided as a source data file.

Assessment of B/Col NP expression in HEK293T transfected cells was confirmed by Western blotting (Fig. 1d). B/Col M2 expression was not assessed because no influenza B virus anti-M2 antibody was commercially available.

### Nucleoside-modified mRNA-LNP vaccination elicits robust humoral immune responses

After determining efficient expression of vaccine antigens in transfected HEK293T cells, we next investigated the titers and functionality of serum antibodies produced 28 days after a prime-only (Fig. 2a) or 28 days after a prime-boost vaccination regimen, where the prime and boost were separated by 28 days (Fig. 2b). Mice were vaccinated intradermally (I.D.) with nucleoside-modified mRNA-LNPs encoding different IBV antigens (monovalent or pentavalent formulation) or an irrelevant formulation encoding firefly luciferase. Antibody titers determined in enzyme-linked immunosorbent assays (ELISAs) following a prime-only or prime-boost immunizations indicated that the vaccines elicited potent antigen-specific antibodies, with similar results observed when most of the constructs were administered individually or in combination (Fig. 3). Antibody responses were amplified after two immunizations compared to the administration of a single vaccine dose. When assessing the antibody titers to the NP, we noted that the individual NP mRNA-LNP construct elicited more robust antibody responses in both the prime-only and prime-boost sera compared to the pentavalent formulation (Fig. 3e). Our results also demonstrate minimal differences in ELISA titers between the pentavalent groups, where mice were given 5 or 1 µg of each antigen.

To further assess the vaccine-elicited antibodies, a multicycle neutralization assay was performed using B/Colorado/06/2017 (V) (Fig. 4a–c) or B/Phuket/3073/2013 (Y) (Fig. 4d–f) virus. The lineage-matched HA and pentavalent formulations were found to elicit high neutralizing titers in the prime-only sera (Fig. 4a, d). Sera from B/Col NA-vaccinated mice were only found to neutralize the B/Victoria/2/1987-like virus (Fig. 4a), not the B/Yamagata/16/1988-like virus (Fig. 4d). Sera from the B/Col NP, B/Col M2, and non-lineage matched HA vaccination groups did not show neutralization in the multicycle neutralization assay (Fig. 4a, d). After determining that sera from B/Col NA-vaccinated mice failed to neutralize the B/Yamagata/16/1988-like

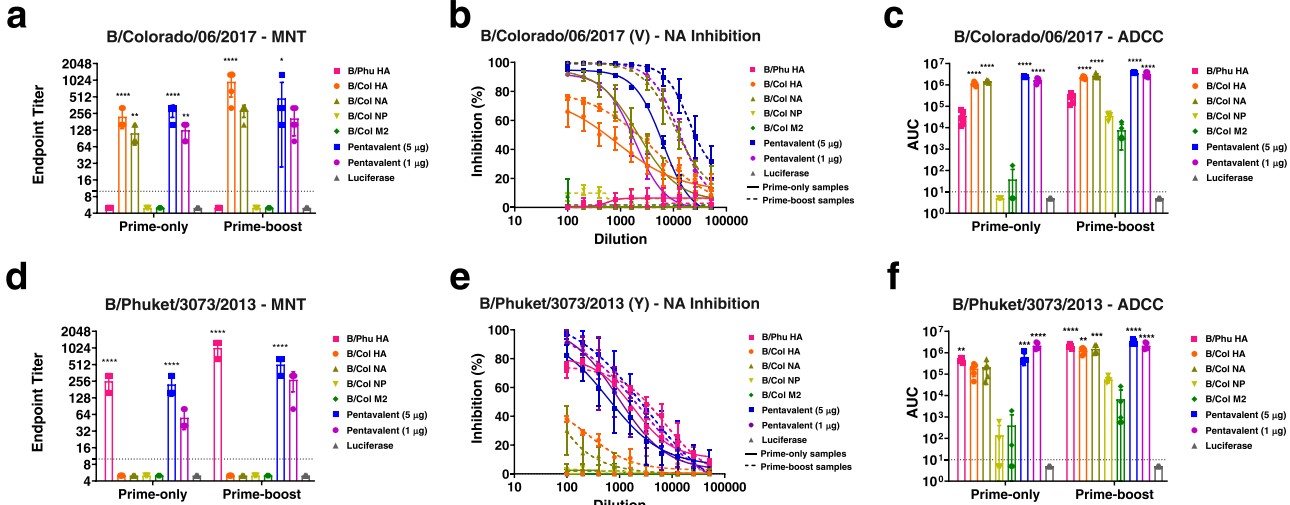

**Fig. 4 | Functional characterization of antibodies against B/Colorado/06/2017 (V) and B/Phuket/3073/2013 (Y).** Mice were vaccinated and sera were collected as described in Fig. 2. Functional characterization of antibodies in the prime-only and prime-boost sera of mice vaccinated with mRNA-LNPs was performed in micro-neutralization (MNT) ($n = 5$) (p-values left to right <0.0001, 0.0047, <0.0001, 0.0011, <0.0001, 0.0154) (**a**), NAI ($n = 3$) (**b**), or ADCC reporter ($n = 5$) ($p < 0.0001$ for all comparisons) (**c**) assay against B/Colorado/06/2017 (V) influenza virus. Functional characterization of antibodies in the prime-only and prime-boost sera of mice vaccinated with mRNA-LNPs was performed in MNT ($n = 5$) (p-values left to right <0.0001, <0.0001, <0.0001, <0.0001) (**d**), NAI ($n = 3$) (**e**), or ADCC reporter ($n = 5$) (p-values left to right 0.0059, 0.0006, <0.0001, <0.0001, 0.0017, 0.0002, <0.0001, 0.0001) (**f**) assay against B/Phuket/3073/2013 (Y) influenza virus. In **b** and **e**, solid lines show NAI values obtained from prime-only sera, dashed lines show NAI values obtained from prime-boost sera. Sera from 3 to 5 randomly selected animals from each vaccination group was assessed in each assay. For **a**, **c**, and **f** the dotted lines indicate the limit of detection. For **a**, **c**, **d** and **f**, significance was assessed using a one-way ANOVA and groups were compared to the luciferase control group at the prime-only or prime-boost time point. *$p < 0.05$, **$p < 0.01$, ***$p < 0.001$, ****$p < 0.0001$. For **a**, **c**, **d** and **f**, bars represent the group mean and error bars represent SD. Each symbol represents an individual animal. For **b** and **e**, data are presented as group mean at each sera dilution and error bars represent SD. Source data are provided as a source data file.

virus, we were interested in establishing if this translated to inhibition of a B/Yamagata/16/1988-like NA. Here, NA-inhibiting antibodies were enumerated in sera from mice vaccinated once against a B/Victoria/2/1987-like (B/Colorado/06/2017) and B/Yamagata/16/1988-like (B/Phuket /3073/2013) virus. Our data indicated that mice vaccinated with B/Col NA had strong inhibition of the B/Colorado/06/2017 NA (Fig. 4b), but not the B/Phuket/3073/2013 NA (Fig. 4e). Our data also showed that mice vaccinated with formulations that contained the lineage-matched HA were able to exhibit NA inhibition (NAI). This could be due to steric hindrance mediated by anti-HA antibodies[17] or by anti-carbohydrate antibodies[18]. To assess the ability of serum antibodies to elicit Fc-mediated effector functions, a murine antibody-dependent cell-mediated cytotoxicity (ADCC) reporter assay was performed using a B/Victoria/2/1987-like (B/Colorado/06/2017) and B/Yamagata/16/88-like (B/Phuket/3073/2013) virus. Sera from mice vaccinated with all formulations, except the B/Col NP and B/Col M2, displayed strong activity in the ADCC reporter assays at the prime-only time point against B/Colorado/06/2017 (Fig. 4c) and B/Phuket/3073/2013 (Fig. 4f).

We also assessed functional antibody responses following prime-boost vaccination. Our results indicate increases in neutralization titer for vaccinated mice that were found to neutralize B/Colorado/06/2017 (V) (Fig. 4a) or B/Phuket/3073/2013 (Y) (Fig. 4d) following a prime-only immunization. Increased NAI titers were also observed for B/Colorado/06/2017 (Fig. 4b) and B/Phuket/3073/2013 (Fig. 4e) viruses. We also observed NAI from the sera of B/Col HA- and B/Col NA-vaccinated mice against B/Phuket/3073/2013 (Y) (Fig. 4e), which was not observed at the prime-only time point. Mice from all vaccination groups, except the luciferase control group, were found to have Fc effector reporter activity against B/Colorado/06/2017 (V) and B/Phuket/3073/2013 (Y) at the prime-boost time point when assessed in an ADCC reporter assay (Fig. 4c, f).

Overall, the antibodies elicited by nucleoside-modified influenza B virus antigen-encoding mRNA-LNP vaccines were antigen-specific and showed functionality in multiple assays.

## Nucleoside-modified mRNA-LNP vaccination elicits antigen-specific cellular immune responses

To investigate the induction of cellular immune responses elicited by vaccination with nucleoside-modified mRNA-LNPs encoding IBV antigens, CD4+ and CD8+ T cell responses were evaluated. In line with our previous studies[13,19], we found that the B/Phu HA, B/Col HA, NA and NP monovalent vaccines were able to elicit both antigen-specific CD4+ and CD8+ T cell responses in mice 10 days after a single I.D. immunization with 5 μg mRNA-LNPs (Fig. 5 and Supplementary Fig. 1). Of note, NP-specific IFN-γ+CD8+ T cell responses were particularly robust raising the possibility that these T cells may contribute to vaccine-induced protective immune responses.

## Nucleoside-modified mRNA-LNP-vaccinated mice are protected from challenge with a broad panel of IBVs

To assess the potential of a prime-only vaccination approach to provide protection from IBVs, challenge with a broad panel of IBVs was assessed. IBV strains were selected to represent ancestral (B/Lee/1940), B/Yamagata/16/1988-like (B/Florida/04/2006 and B/New York City/PV00094/2017) and B/Victoria/2/1987-like (B/Malaysia/2506/2004 and B/New York City/PV01181/2018) IBVs (Fig. 2a). The amino acid similarity for vaccine antigens and challenge strains are detailed in Table 1. Mice were challenged with 5mLD$_{50}$ of each respective virus and weight loss was monitored for morbidity and mortality over a 14-day time course (Fig. 6). Our results indicate that mice vaccinated with the B/Phu HA, B/Col HA, or B/Col NA monovalent formulations and pentavalent formulations were protected from challenge with minimal morbidity observed. Unsurprisingly, challenge with B/Victoria/2/1987-like viruses resulted in more morbidity in B/Phu HA (a B/Yamagata/16/1988-like HA) vaccinated mice when compared to mice vaccinated with the B/Col HA (a B/Victoria/2/1987-like HA). The reverse was observed upon challenge with B/Yamagata/16/1988-like viruses. Mice vaccinated with the B/Col NP lost a significant amount of body weight, but these mice were mostly

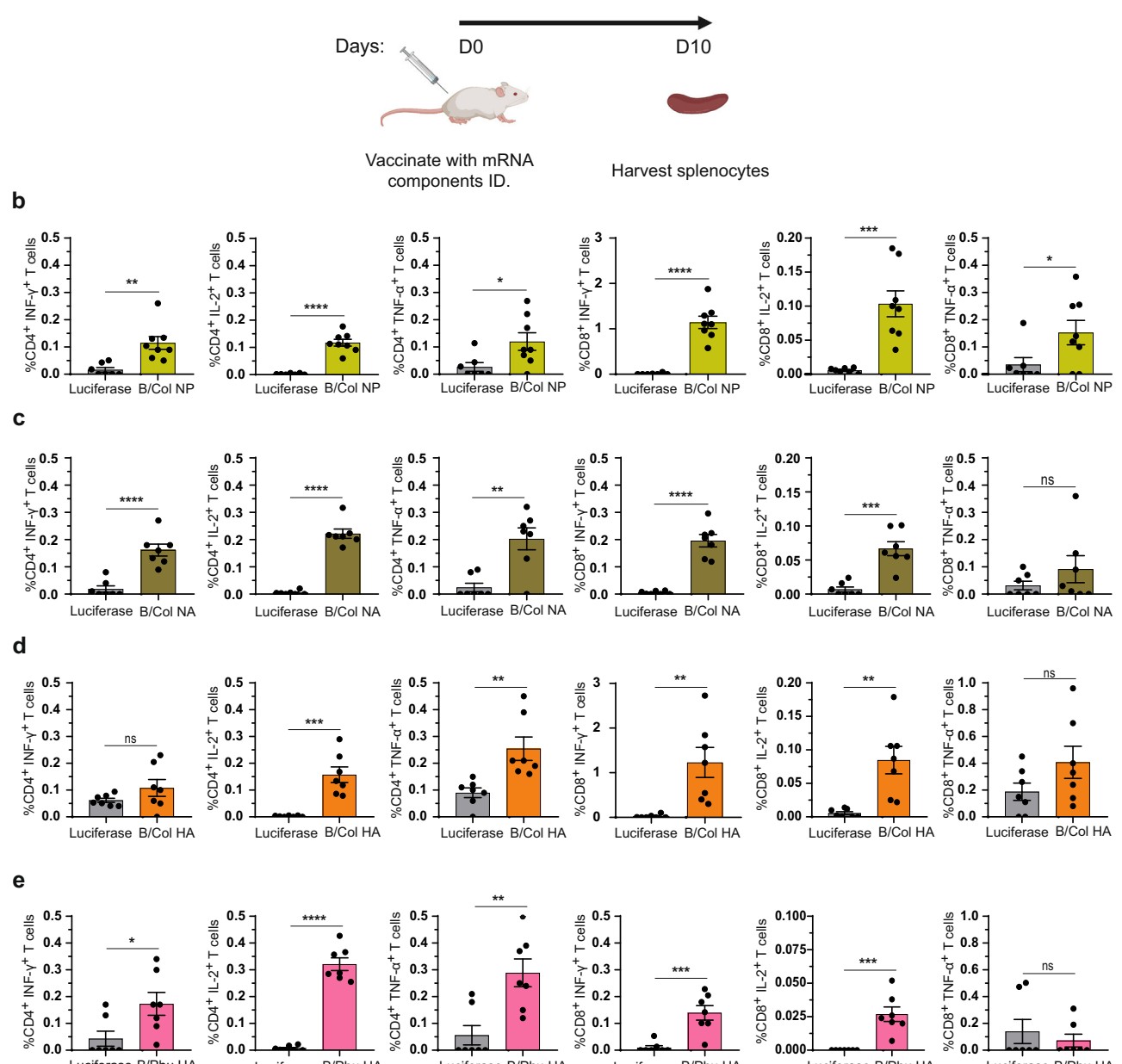

**Fig. 5 | Cellular immune responses induced by IBV mRNA-LNP vaccines.** Mice were vaccinated I.D. with a single dose of 5 µg of NP, NA, HA or control luciferase mRNA-LNPs. Splenocytes collected from immunized animals 10 days after immunization were stimulated with NP or NA or HA overlapping peptide pools, and cytokine production by CD4+ and CD8+ T cells was assessed by flow cytometry (**a**). Percentages of B/Col NP- (*p*-values left to right 0.0023, <0.0001, 0.028, <0.0001, 0.0004, 0.0478) (**b**), B/Col NA- (*p*-values left to right <0.0001, <0.0001, 0.0014, <0.0001, 0.0001, 0.2702) (**c**), B/Col HA- (*p*-values left to right 0.1794, 0.0002, 0.0047, 0.0037, 0.0023, 0.1322) (**d**) and B/Phu HA- (*p*-values left to right 0.026, <0.0001, 0.0031, 0.0006, 0.0003, 0.5177) (**e**) specific CD4+ and CD8+ T cells producing IFN-γ, IL-2 and TNF-α are shown. Each symbol represents one animal and error is shown as SEM (*n* = 8 mice per group for B/Col NP and *n* = 7 for B/Col NA, B/Col HA, B/Phu HA and luciferase). Statistical analysis: two-tailed unpaired *t*-test, ∗*p* < 0.05, ∗∗*p* < 0.01, ∗∗∗*p* < 0.001, ∗∗∗*p* < 0.0001. Source data are provided as a source data file.

protected from challenge against all IBVs tested, except the B/New York City/PV00094/2017 (Y) IBV where almost all B/Col NP-vaccinated mice succumbed to infection (Fig. 6e, j). Mice vaccinated with the B/Col M2 or the luciferase control mRNA-LNP formulations were susceptible to challenge with all IBVs.

Following these studies, we compared the performance of our IBV mRNA-LNP vaccines with a conventional vaccine platform (trivalent influenza virus vaccine, TIV, Fluzone, 2006–2007) in a prime-only vaccination regimen (Supplementary Fig. 2). We found that the pentavalent mRNA-LNPs, either at the 5 µg or the 1 µg dose, outperformed

the TIV. High antibody titers were detected against B/Phu HA (Y), B/Col HA (V), or B/Col NA following administration of the pentavalent formulations, while TIV only induced detectable antibodies against B/Col HA (V) (Supplementary Fig. 2a–d). Four weeks after immunization, animals were I.N. challenged with the B/Malaysia/2506/2004 (V) strain, one of the components of the TIV vaccine (Supplementary Fig. 2e, f). In line with our previous observations (Fig. 6), luciferase controls and M2-immunized mice displayed severe morbidity and succumbed to infection, while mice in all the other groups survived challenge. Notably, TIV-vaccinated mice survived challenge (Supplementary

**Table 1 | Amino acid similarity (%) between vaccine antigens and challenge strains**

| Viruses | Vaccine antigen | | | | |
|---|---|---|---|---|---|
| | B/Phu HA | B/Col HA | B/Col NA | B/Col NP | B/Col M2 |
| **B/Lee/1940** | 91.27 | 91.94 | 88.84 | 95.71 | 81.65 |
| **B/Malaysia/2506/2004 (V)** | 92.98 | 97.94 | 96.35 | 98.21 | 97.25 |
| **B/Colorado/06/2017 (V)** | 92.28 | 100 | 100 | 100 | 100 |
| **B/New York City/PV01181/2018 (V)** | 92.62 | 99.66 | 99.14 | 99.82 | 100 |
| **B/Florida/04/2006 (Y)** | 98.29 | 92.62 | 94.85 | | |
| **B/Phuket/3073/2013 (Y)** | 100 | 92.28 | 93.78 | 99.20 | 91.74 |
| **B/New York City/PV00094/2017 (Y)** | 99.66 | 92.28 | 93.13 | 99.29 | 89.91 |

Amino acid sequences from vaccine antigens (B/Phu HA, B/Col HA, B/Col NA, B/Col NP or B/Col M2) were aligned to amino acid sequences of proteins from influenza virus strains used in this study (B/Lee/1940, B/Malaysia/2506/2004 (V), B/Colorado/06/2017 (V), B/New York City/PV01181/2018 (V), B/Florida/04/06 (Y), B/Phuket/3073/2013 (Y) or B/New York City/PV00094/2017 (Y) using the Clustal Omega multiple sequence alignment tool. Amino acid sequences of NP and M2 from B/Florida/04/2006 (Y) were not assessed. The percent (%) amino acid identity was determined using the computed percent identity matrix and examined for each challenge virus.

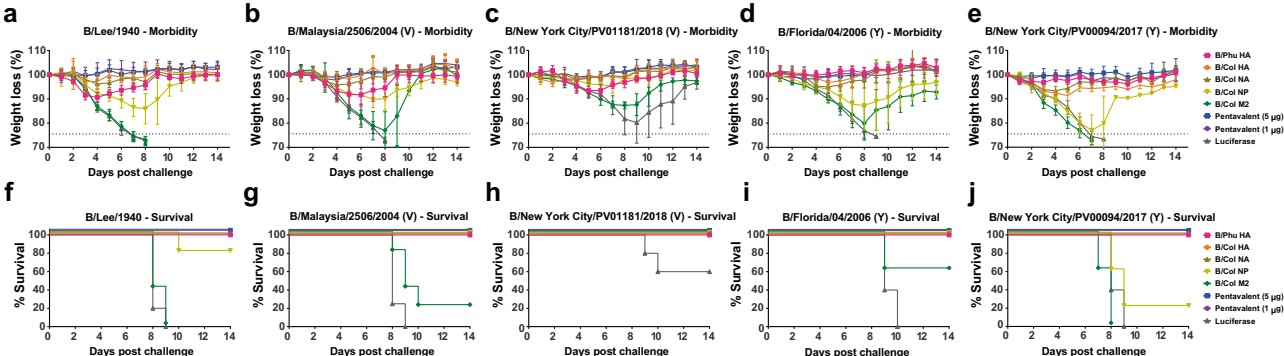

**Fig. 6 | Morbidity and survival graphs for mRNA-LNP-vaccinated mice following I.N. challenge with ancestral through recent IBV strains.** Morbidity of mice vaccinated with a single dose of mRNA-LNPs containing monovalent or pentavalent formulations following I.N. challenge with $5mLD_{50}$ of B/Lee/1940 ($n = 5$) (**a**) B/Malaysia/2506/2004 (V) ($n = 5$) (**b**), B/New York City/PV01181/2018 (V) ($n = 5$ for all groups except M2 where one mouse was found dead on day 1) (**c**), B/Florida/04/2006 (Y) ($n = 5$) (**d**), or B/New York City/PV00094/2017 (Y) ($n = 4$) (**e**) influenza viruses 4 weeks after immunization. Data are shown as the group mean and error bars represent SD. Survival of mice vaccinated with a single dose of mRNA-LNPs containing monovalent or pentavalent formulations following I.N. challenge with $5mLD_{50}$ of B/Lee/1940 (**f**), B/Malaysia/2506/2004 (V) (**g**), B/New York City/PV01181/2018 (V) (**h**), B/Florida/04/2006 (Y) (**i**), or B/New York City/PV00094/2017 (Y) (**j**) influenza viruses 4 weeks after immunization. Source data are provided as a source data file.

Fig. 2f), but displayed marked weight loss and did not return to their starting weights by the end of the observation period (Supplementary Fig. 2e).

We also investigated protection against B/Malaysia/2506/2004 (V) challenge in prime-boost vaccinated mice to assess for further improvements in the prime-only vaccination regimen (Fig. 2b). Similar to the prime-only challenge studies, we found that mice vaccinated with B/Col HA, B/Col NA, and the pentavalent formulations were protected from challenge, with minimal weight loss observed (Supplementary Fig. 3). We also observed improvements in weight loss in mice vaccinated with the B/Col NP, and B/Col M2 mRNA-LNPs when compared to the prime-only vaccination. No improvements in weight loss in mice vaccinated with the B/Phu HA mRNA-LNP were observed when compared to the prime-only vaccination. All mice vaccinated with the luciferase control mRNA-LNP succumbed to infection.

Next, we assessed control of virus titers mediated by these vaccines, as some formulations induced neutralizing antibody responses whereas others did not. We infected vaccinated mice with $1mLD_{50}$ of B/New York City/PV01181/2018 (V) (Fig. 7a) or B/New York City/PV00094/2017 (Y) (Fig. 7b) viruses and harvested lungs at 3 or 6 days post infection (dpi) and measured viral titers in the lung homogenates. We observed that mice vaccinated with B/Col HA, B/Col NA, and the pentavalent formulations had reduced or no measurable virus titers following challenge with B/New York City/PV01181/2018 (V) at 3 dpi, and no virus was detected at 6 dpi (Fig. 7a). In contrast, mice

vaccinated with B/Phu HA, B/Col NP, B/Col M2, and luciferase control had measurable virus titers at 3 and 6 dpi. Mice vaccinated with B/Phu HA and the pentavalent formulations had reduced virus titers following challenge with B/New York City/PV00094/2017 (Y) at 3 dpi, and no virus was detected at 6 dpi (Fig. 7b). Mice vaccinated with B/Col HA and B/Col NA had quantifiable virus titers at 3 dpi but no detectable titers at 6 dpi. Mice vaccinated with B/Col NP, B/Col M2, and luciferase control had measurable titers at both 3 and 6 dpi. Our results highlight that 1 μg of the pentavalent formulation resulted in sterilizing protection when compared to the monovalent formulations. These data also suggest that neutralizing antibodies are required for complete control of virus replication.

### Dose de-escalation with the pentavalent formulation shows protection in the nanogram range after a single vaccination

After observing the broad protection elicited by the pentavalent mRNA-LNP vaccine formulation against ancestral through recent IBVs in a prime-only vaccination study (Fig. 6), we next determined the minimal dose at which the pentavalent formulation was protective. Mice were vaccinated with 10-fold decreasing doses of the influenza B virus pentavalent formulation or the luciferase control. Twenty-eight days after vaccine administration, mice were bled, and sera were analyzed by ELISA against cells infected with B/Colorado/06/2017 (V) (Fig. 8a). Measurable antibody responses were detected for mice vaccinated with 5 μg, 0.5 μg, or 0.05 μg of each antigen in the

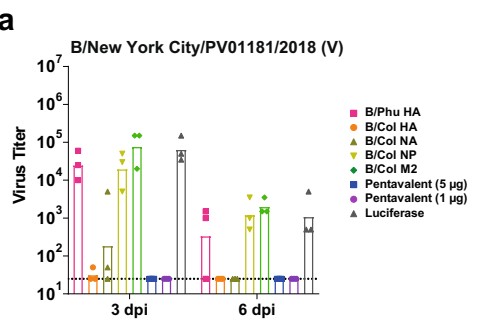

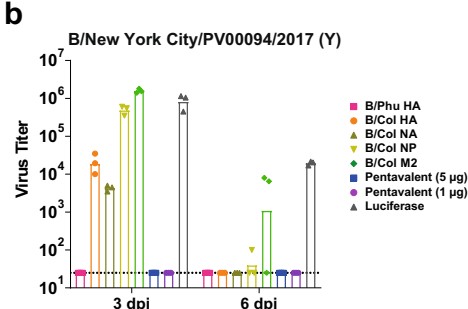

**Fig. 7 | A single immunization with IBV pentavalent mRNA-LNP vaccine prevents virus replication.** Mice were I.D. vaccinated once with 5 μg of monovalent mRNA-LNPs or with 5 μg/antigen or 1 μg/antigen of the pentavalent mRNA-LNP formulation. Control animals received 5 μg of luciferase mRNA-LNP. Virus titers in the lung homogenate of mRNA-LNP-vaccinated mice at 3 and 6 dpi following I.N. challenge with 1mLD$_{50}$ of B/New York City/PV01181/2018 (V) (n = 3 mice/group) (**a**).

Virus titers in the lung homogenate of mRNA-LNP-vaccinated mice at 3 and 6 dpi following I.N. challenge with 1mLD$_{50}$ of B/New York City/PV00094/2017 (Y) (n = 3 mice/group) (**b**). Each symbol represents one animal and the average is presented as the group mean. The dotted lines indicate the limit of detection. Source data are provided as a source data file.

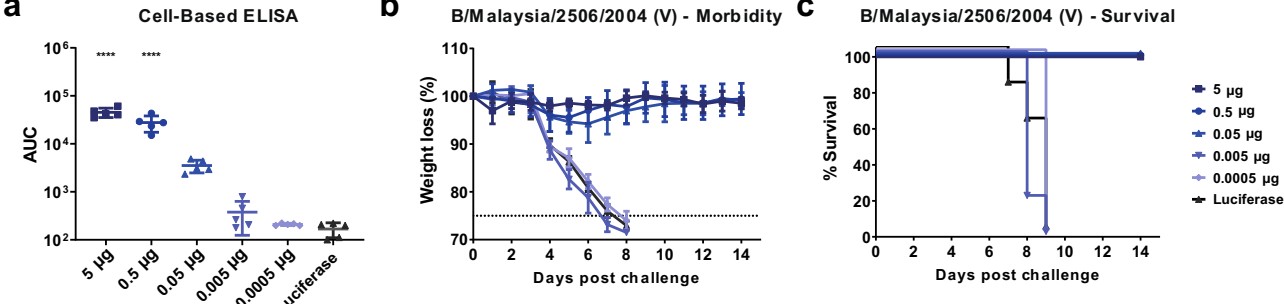

**Fig. 8 | A single immunization with the pentavalent IBV mRNA-LNP vaccine protects mice in the nanogram dose range.** Mice were I.D. vaccinated once with 5, 0.5, 0.05, 0.005, or 0.0005 μg of each antigen in the pentavalent mRNA-LNP formulation. Four weeks later sera were collected and antibody responses towards B/Colorado/06/2017 (V) influenza virus-infected cells were assessed (p-values from left to right <0.0001, <0.0001) (**a**). Each symbol represents one animal, and the bar represents the mean. Mice were then I.N. infected with

5mLD$_{50}$ of B/Malaysia/2506/04 (V) influenza virus and morbidity (**b**) and survival (**c**) were monitored for 14 dpi. Data are shown as individual data points and the group average presented as the group mean (n = 5 per group). Significance was assessed using a one-way ANOVA and groups were compared to the luciferase control group at each time point. The dotted line in **b** indicates the maximum body weight loss (25%) for the experiment. Source data are provided as a source data file.

pentavalent formulation. Limited to no antibody responses were detected in mice vaccinated with 0.005 μg or 0.0005 μg of the pentavalent formulation. Mice were then challenged with 5mLD$_{50}$ of B/Malaysia/2506/2004 (V) virus and weight loss and survival were monitored for 14 days post challenge (Fig. 8b, c). All vaccinated mice were protected from infection at the 5, 0.5, and 0.05 μg doses, with limited morbidity and no mortality observed. All mice vaccinated with 0.005 or 0.0005 μg of the pentavalent vaccine or the luciferase control vaccine succumbed to infection. In summary, vaccination with a single low dose of 0.05 μg/antigen of the pentavalent mRNA-LNP formulation can protect animals from morbidity and mortality against lethal viral challenge.

**Antibodies that target the influenza virus surface glycoproteins reduce morbidity upon serum passive transfer challenge studies**
To determine if antibodies induced by mRNA-LNP vaccination could reduce morbidity following IBV infection, a passive transfer experiment was conducted. Mice were vaccinated twice with 5 μg of mRNA-LNP vaccines (monovalent and pentavalent formulations) with 28-day intervals between administrations to generate strong immune responses. Four weeks after the boost, a terminal bleed was performed to collect sera. Sera from the terminal bleeds were tested against B/Colorado/06/2017 (V) virus-infected cells by ELISA and showed that sera from mice vaccinated with monovalent and pentavalent formulations were highly reactive in this assay (Fig. 9a). This serum was

then pooled within groups and transferred into naive mice through intraperitoneal administration. Two hours after transfer, sera from recipient mice were harvested and subsequently tested against B/Colorado/06/2017 (V) virus-infected cells by cell-based ELISA (Fig. 9b). The post-transfer sera reacted similarly to the pre-transfer sera, although an expected reduced reactivity was observed. Animals were then challenged with 5mLD$_{50}$ of B/New York City/PV01181/2018 (V) virus and weight loss was monitored for 14 days post challenge. Animals that received serum from mice vaccinated with the pentavalent formulations, the B/Col HA, or the B/Col NA were protected from challenge with no weight loss (Fig. 9c). Animals that received serum from mice vaccinated with the B/Phu HA were also protected from challenge, although a maximum loss of 8.5% of the original body weight was observed. Mice that received sera from B/Col NP-, B/Col M2-, or luciferase control-immunized donors showed severe morbidity (Fig. 9c), with this data suggesting that only antibodies that target the surface glycoproteins prevent morbidity in a passive transfer experiment.

## Discussion
Nucleoside-modified mRNA-LNP vaccines have emerged as a powerful vaccine platform for the control of infectious diseases[20]. Our previous works investigating the use of the nucleoside-modified mRNA-LNP vaccine platform for the development of a broadly protective influenza virus vaccine were successful. In those studies we used mRNA-encoded

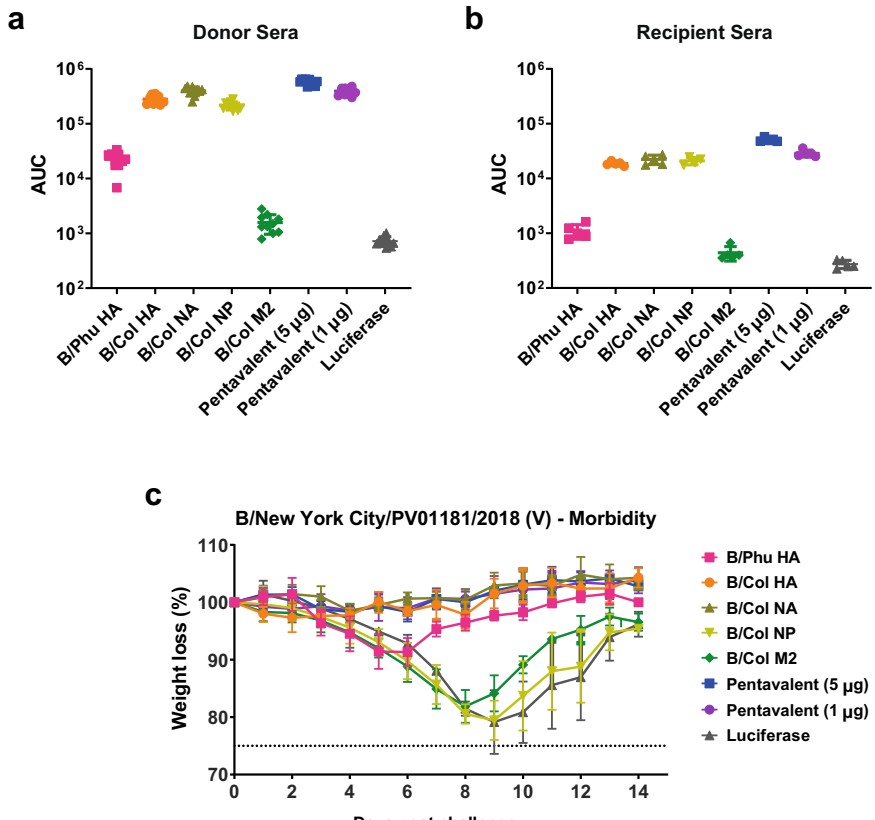

**Fig. 9 | Humoral protection from IBV challenge is afforded by antibodies that target glycoproteins.** Mice were vaccinated twice (with a 4-week interval) I.D. with 5 μg of mRNA-LNPs. Sera were collected from vaccinated animals 4 weeks after the boost and then assessed for antibody reactivity towards B/Colorado/ 06/2017 (V) influenza virus-infected cells ($n = 10$ mice/group) (**a**). Sera from vaccinated mice were transferred into naïve mice. Two hours after the transfer, sera from naïve recipients of passive sera were assessed for antibody reactivity towards B/Colorado/06/2017 (V) influenza virus-infected cells ($n = 5$ mice/ group) (**b**). For a and b bars represent the mean of each group and each point represents an individual animal. Error bars are representative of SD. One mouse was excluded from the NP group due to failed serum transfer. Naïve mice were then I.N. challenged with $5 mLD_{50}$ of B/New York City/PV01181/2018 (V) influenza virus and weight loss was monitored for 14 days. Data are shown as mean and error bars represent SD ($n = 5$ per group) (**c**). The dotted line in **c** indicates the maximum body weight loss (25%) for the experiment. Source data are provided as a source data file.

influenza virus immunogens (A/Brisbane/59/2007 H1 headless HA construct, a membrane-bound A/Michigan/45/2015 N1 NA, A/Michigan/45/2015 NP and A/Michigan/45/2015 M2) to provide broad protection against group 1 challenge viruses (based on the HA)[13]. In a subsequent study we demonstrated that introduction of modifications into the mRNA-encoded influenza virus antigens could improve the safety and immunogenicity of nucleoside-modified mRNA-LNP vaccines[14]. In this current work, we aimed to develop a broadly protective IBV vaccine that targeted multiple antigens. We compared the immunogenicity and protective efficacy of singular mRNA-LNP constructs encoding HA of the B/Victoria/2/1987-like or B/Yamagata/16/ 1988-like lineage, NA, NP, or M2 from the B/Victoria/2/1987-like lineage to a pentavalent mRNA-LNP formulation, which encoded all 5 IBV antigens in mice. Our study proves that the pentavalent mRNA-LNP vaccine induces broadly protective immune responses and outperforms the monovalent constructs providing the basis for further investigation of this vaccination regimen. Of note, here we used naïve mice to evaluate our vaccine formulations. As prior exposure to influenza virus alters vaccine-induced immune responses in humans[21], future studies should investigate the immunogenicity and protective efficacy of the pentavalent mRNA-LNP vaccine in influenza-exposed animal models to mimic pre-existing immunity present in humans.

In this work we vaccinated mice with nucleoside-modified mRNA-LNPs containing 5 μg of B/Phu HA or 5 μg of B/Col HA, NA, NP, or M2. We found that most of these monovalent mRNA-LNP vaccine formulations protected mice from challenge with all or most IBVs

after administration of a single vaccine dose. Moreover, we demonstrated that a pentavalent IBV mRNA-LNP formulation outperformed all individual components. Minimal to no weight loss was observed in mice vaccinated with 5 μg or 1 μg of each antigen in the pentavalent formulations upon challenge with each IBV when compared to the monovalent formulations. Impressively, no virus was detected in the lungs of mice vaccinated with the pentavalent formulation following challenge with the mismatched IBV lineage when compared to the monovalent components. The pentavalent formulation was also able to protect mice from challenge with B/ Malaysia/2506/2004 (V) when mice were vaccinated with only 50 ng of each antigen. Fifty μg of HA mRNA-LNPs was recently described by Moderna to induce neutralizing antibody responses in vaccinated humans and the dose we used in our study may be comparable to this study[22]. The only difference we found between the pentavalent and monovalent formulations was a reduction in the amount of anti-NP antibodies in the pentavalent formulations when compared to the monovalent formulations, a reduction that did not impact any of the functional assays performed in this study. When we assessed the antibody responses in mice vaccinated with the pentavalent formulations, we found modest improvements in functional antibody responses in the assays that we tested (MNT, NAI, or ADCC reporter assay), suggesting that vaccine-induced antibody functionality is not affected by the number of antigens in the pentavalent mRNA-LNPs. The modest improvement we observed in the pentavalent formulations may be attributed to the

combined multiantigen-targeted immune response as described below for each antigen.

Numerous studies have indicated that broadly protective responses against IAVs have been successful by utilizing vaccination regimens that target the conserved HA stalk domain[23–26]. Headless HA constructs for group 1 and group 2 IAVs have allowed the direct targeting of the immuno-subdominant HA stalk[27–29]; however, headless HA constructs have not been developed or described for IBV HAs (although mosaic HA constructs, which redirect the immune response towards the immuno-subdominant conserved epitopes of the HA via sequential immunization are being developed[7,30,31]). Therefore, in our studies, we used full-length HA constructs from either a B/Yamagata/16/1988-like or B/Victoria/2/1987-like virus, like in current QIVs. We showed that vaccination with a HA of either lineage resulted in protection against challenge with either a B/Yamagata/16/88-like or B/Victoria/2/87-like virus, although increased morbidity was observed when mice were vaccinated with non-lineage matched HAs. Interestingly, we showed that non-lineage matched anti-HA antibodies were non-neutralizing and did not aid in the control of virus titers (although virus titers were somewhat reduced). The protection afforded by non-lineage matched antibodies is mostly mediated through effector functions. Cross-lineage protection has previously been observed in non-lineage matched anti-HA mAbs that bound the HA stalk of IBVs and induced Fc-effector functions[32]. Indeed, both IBV HA mRNA-LNPs induced cross-protection, suggesting that vaccine-induced HA stalk-specific antibodies contributed to the protective immune responses. Of note, cross-protection could potentially be achieved by utilizing a single mosaic IBV HA construct, where the antigenic sites in the immune-dominant HA head are replaced by exotic IAV sequences[7].

Unlike IBV HAs, IBV NAs have not separated into distinct lineages, although recent antigenic analysis has shown that the NA of B/Yamagata/16/88-like viruses is under greater antigenic selection[3]. In line with previous studies[5], our work shows that vaccination with a B/Victoria/2/87-like NA from B/Colorado/06/2017 (V) leads to protection against challenge with both B/Yamagata/16/88-like and B/Victoria/2/87-like viruses. Similarly to the HA monovalent mRNA-LNP formulations, we found that neutralizing antibodies, NA-inhibiting antibodies, and reduced virus titers were only observed in NA-vaccinated mice in the context of B/Victoria/2/1987-like viruses. The use of a B/Yamagata/16/1988-like virus resulted in no measurable NAI activity and detectable virus titers in the lung upon challenge in mice vaccinated only once (although the virus titers were reduced). Despite this finding, Fc effector functions were detected against both viruses in the ADCC reporter assay. The lack of antibodies with cross-lineage NAI activity is interesting, as previous vaccination studies have identified such antibodies[5]. In this study we only assessed one mismatched virus (B/Phuket/3073/2013 (Y)), and these observations may not be true for all B/Yamagata/16/1988-like viruses. Of note, these experiments utilized a prime-boost vaccination regimen. Whilst we mostly performed prime-only experiments, the use of a prime-boost vaccination regimen resulted in only minimal amount of NA-inhibiting antibodies, and perhaps antigenic differentiation of recent IBV NAs is leading to reduced targeting of the NA active site[3].

The use of NP and M2 IBV proteins as vaccine antigens is limited and most previous work has focused on assessing NP and M2 IAV proteins[11–13,33–36]. These studies showed that NP vaccination led to protection via cellular immunity and conserved epitopes in M2 could induce heterosubtypic immunity. One study showed that intranasal vaccination with an adenovirus vector expressing an IBV NP resulted in robust cellular immunity and cross-lineage protection[37]. Our work indicates that vaccination with an IBV NP leads to robust antibody responses and NP-vaccinated mice are mostly protected from influenza virus challenge, although weight loss was observed in animals after infection with each challenge virus. As expected, vaccination with the NP antigen did not result in the induction of neutralizing antibodies, although we observed low functioning antibody responses in the ADCC reporter assay in sera taken after 2 immunizations. The transient expression of NP on the surface of influenza virus-infected cells has been observed previously and this transient expression may be sufficient for the generation of effector antibodies[38]. It is also possible that the NP protein made from the mRNA inside the cells is released upon cell death and is then able to induce an antibody response. NP-specific antibodies isolated from influenza-infected patients have been recently associated with Fc effector functions[39]. However, passive transfer of these antibodies into a murine influenza challenge model resulted in no protection, complementing our passive transfer studies. Overall, these data suggest that the protection afforded by targeting the NP is not provided by antibodies. NP contains many epitopes targeted by T cells, and targeting these epitopes can be highly protective in the mouse model[40,41] and both CD8+ and CD4+ T cell responses correlate with reduced infection in humans[42,43]. Here, we found that the B/Col NP mRNA-LNP vaccine induced both antigen-specific CD4+ and CD8+ T cell responses after administration of a single dose. Particularly, IFN-γ+CD8+ T cell responses were robust and it is highly likely that the protection observed in NP mRNA-LNP-vaccinated animals was afforded by T cell-mediated immunity, although this was not directly investigated in our experiments.

The IBV M2 provided no protection in the prime-only-vaccinated mice and measurable antibody responses were only observed in sera from prime-boost-vaccinated mice. However, some protection was observed in animals that received two vaccine doses. The IBV M2 is lined with polar residues that are believed to reduce the hydrophobic interactions of adamantanes[44,45], an antiviral drug against IAVs. These polar residues may prevent antibody binding and may be the reason we observe a blunted response in IBV M2-vaccinated mice.

Taken together, our work supports the development of nucleoside-modified mRNA-LNP vaccines for the control of influenza virus, and importantly those that target IBVs. As IBVs lack an animal reservoir, vaccines that target IBV infections could, with sufficient vaccine uptake, limit virus spread of IBVs in the human population and could ultimately lead to the eradication of IBVs. These data encourage further exploration of multivalent mRNA-based influenza virus vaccines that induce broadly protective immune responses through targeting multiple antigens.

## Methods

### Ethics statement

The investigators faithfully adhered to the "Guide for the Care and Use of Laboratory Animals" by the Committee on Care of Laboratory Animal Resources Commission on Life Sciences, National Research Council. Mouse studies were conducted under protocols approved by the Institutional Animal Care and Use Committees (IACUC) of the University of Pennsylvania (UPenn) and the Icahn School of Medicine at Mount Sinai (ISMMS). All animals were housed and cared for according to local, state, and federal policies in an Association for Assessment and Accreditation of Laboratory Animal Care International (AAALAC)-accredited facility.

### Viruses & cells

Sf9 cells (CRL-1711, ATCC) for baculovirus rescue were grown in *Trichoplusia ni* medium-formulation Hink insect cell medium (TNM-FH, Gemini Bioproducts) supplemented with 10% fetal bovine serum (FBS; Sigma) and penicillin (100 U/mL)-streptomycin (100 μg/mL) solution (Gibco). BTI-*TN*-5B1-4 (High Five) (B85502; ThermoFisher) cells for protein expression were grown in serum-free SFX medium (HyClone) supplemented with penicillin (100 U/mL)-streptomycin (100 μg/mL) solution. Madin Darby Canine Kidney (MDCK) (CCL-34, ATCC) and Human Embryonic Kidney (HEK) 293 T (CRL-3216, ATCC) cells were grown in Dulbecco's Modified Eagle's Medium (DMEM) supplemented with 10% FBS and penicillin (100 U/mL)-streptomycin (100 μg/mL)

solution. Expi293F cells (ThermoFisher) were grown in Expi293™ Expression Medium (Gibco) in 1 L Erlenmeyer shake flasks (Corning) at 37 °C and 125 RPM in a humidified incubator with 8% CO$_2$.

The IBV strains B/Colorado/06/2017 (V), B/Phuket/3073/2013 (Y), B/Lee/1940, B/Malaysia/2506/2004 (V), B/Florida/04/2006 (Y), B/New York City/PV01181/2018 (V), and B/New York City/PV00094/2017 (Y) were grown in 10-day-old embryonated chicken eggs (Charles River) for 72 h at 33 °C. Eggs were then cooled overnight (O/N) at 4 °C before allantoic fluid was harvested. Harvested allantoic fluid was centrifuged at 4000 × $g$ for 10 min at 4 °C to pellet debris. Virus stocks were then aliquoted and stored at −80 °C prior to determining stock titers via plaque assay.

## Protein production
Recombinant HAs and NAs from B/Colorado/06/2017 and B/Phuket/3073/2013 were expressed in High Five insect cells as previously described[46]. Recombinant NP from B/Colorado/06/2017 was expressed in HEK293F cells as previously described[13,47]. Proteins were purified from the cell culture supernatant via Ni$^{2+}$-nitrilotriacetic acid (Ni-NTA) chromatography[46–48].

## mRNA-LNP production
mRNA-LNP vaccines were produced as previously described[13,14]. Briefly, codon-optimized firefly luciferase, B/Phuket/3073/2013 HA (B/Phu HA), B/Colorado/06/2017 HA (B/Col HA), B/Colorado/06/2017 NA (B/Col NA), B/Colorado/06/2017 NP (B/Col NP), and B/Colorado/06/2017 M2 (B/Col M2) were synthesized (Genscript). Constructs were ligated into mRNA production vectors, vectors were linearized, and a T7-driven in vitro transcription reaction (Megascript, Ambion) was performed to generate mRNA with 101 nucleotide-long poly(A) tails. Capping of mRNA was performed in concert with transcription through addition of a trinucleotide cap1 analog, CleanCap (TriLink), and m1Ψ-5′-triphosphate (TriLink) was incorporated into the reaction instead of UTP. Cellulose-based purification of mRNA was performed as described[49]. mRNAs were then assessed on an agarose gel before storing at −20 °C.

Purified mRNAs were encapsulated into lipid nanoparticle using a self-assembly process where an ethanolic lipid mixture of an ionizable cationic lipid, phosphatidylcholine, cholesterol, and polyethylene glycol-lipid was rapidly combined with an aqueous solution containing mRNA at acidic pH as previously described[50]. The ionizable cationic lipid (pK$_a$ in the range of 6.0-6.5, proprietary to Acuitas Therapeutics) and LNP composition are described in the patent application WO 2017/004143. The average hydrodynamic diameter was ~80 nm with a polydispersity index of 0.02-0.06 as measured by dynamic light scattering using a Zetasizer Nano ZS (Malvern Instruments Ltd, Malvern, UK) and an encapsulation efficiency of ~95% as determined using a Ribogreen assay.

## Cell transfections
Transfection of HEK 293 T cells was performed with TransIT-mRNA (Mirus Bio) according to the manufacturer's instructions: mRNA (0.3 μg) was combined with TransIT-mRNA Reagent (0.34 μL) and Boost Reagent (0.22 μL) in 17 μL of serum-free medium, and the complex was added to 6 × 10$^4$ cells in 183 μL complete medium.

## Western blot analysis
Cell lysates of luciferase or B/Col NP mRNA-transfected and non-transfected HEK 293 T cells were used. Cells were lysed for 1 h on ice in radioimmunoprecipitation assay buffer (RIPA) buffer (Sigma) 24 h after transfection. Samples were combined with 355 mM 2-mercaptoethanol (Bio-Rad) containing 4X Laemmli buffer (Bio-Rad) and were boiled for 5 min and spun at 18000 g for 5 min at room temperature (RT). The samples were separated on a 4–15% precast polyacrylamide Criterion TGX gel (Bio-Rad) for 45 min at 200 V.

Transfer to polyvinylidene fluoride (PVDF) membrane (ThermoFisher) was performed using a semi-dry apparatus (Bio-Rad) at 10 V for 1 h. The membrane was blocked in 5% milk powder-TBST (25 mM Tris, 150 mM NaCl, pH 7.5 + 0.1% Tween-20) for 1.5 h, and then, the membrane was incubated with the anti-NP (mouse monoclonal, 1:10.000, Thermo-Fisher, #MA5-35900) and anti-GAPDH (rabbit monoclonal, 1:5000; Cell Signaling Technology, #14C10) primary antibodies overnight at 4 °C. The membrane was washed with 1X TBST for 30 min and incubated with the horseradish peroxidase (HRP)-conjugated anti-rabbit (goat polyclonal, 1:10.000, ThermoFisher, #65-6120) or anti-mouse (donkey polyclonal, 1:10.000, Jackson Immuno, #715-035-150) secondary antibodies for 1 h at RT. After washing three times for 20 min with 1X TBST at RT, the signal was developed with the HRP Substrate solution (GE Healthcare, Amersham ECL Western Blotting Detection Reagents) and imaged using a ChemiDoc XRS + machine (BioRad).

## Staining and flow cytometric analyses of mRNA-transfected cells and mouse splenocytes
Luciferase, B/Phu HA, B/Col HA, or B/Col NA mRNA-transfected HEK 293 T cells were collected 24 h after transfection and were washed once with 1% FBS (HyClone) in phosphate buffered saline (PBS) (Corning, DPBS 1X (without calcium and magnesium)). Next, cells were incubated with 7.5 μg/ml anti-NA (1G05[9]) or 7.5 μg/ml anti-HA (CR8033 for B/Phu HA and CR8059 for B/Col HA) antibodies in V-bottom 96-well plates (Greiner) at 4 °C for 30 min. After washing with 1% FBS in PBS cells were incubated with anti-human AlexaFluor-647 (goat polyclonal, 1:300, ThermoFisher, #A21445) secondary antibody at 4 °C for 30 min in dark. The samples were washed and re-suspended in FACS buffer (1% FBS in PBS). Flow cytometric data were acquired on a BD LSRII flow cytometer. At least 50,000 events for each sample were recorded and data were analyzed with the FlowJo 10 software.

Spleen single-cell suspensions were made in complete Roswell Park Memorial Institute (RPMI) 1640 medium (ATCC). 3 × 10$^6$ cells per sample were stimulated for 6 h at 37 °C and 5% CO2, in the presence of overlapping NP (BEI Resources, NR-19254), NA (BEI Resources, NR-19254) or HA (BEI Resources, NR-18972) peptide pools at 2.5 μg/mL per peptide. GolgiPlug (5 mg/mL; brefeldin A; BD Biosciences) and Golgi-Stop (10 mg/mL; monensin; BD Biosciences) were added to each sample after 1 h of stimulation. Unstimulated samples for each animal were included. A phorbol 12-myristate-13-acetate (10 mg/mL; Sigma) and ionomycin (200 ng/mL; Sigma)-stimulated sample was included as a positive control. After stimulation, cells were washed with PBS and stained for 10 min in dark at 25 °C with the LIVE/DEAD fixable aqua dead cell stain kit (Life Technologies). Samples were incubated in the Fc blocker (Purified Rat Anti-Mouse CD16/CD32) for 10 min in dark at 4 °C and then surface-stained with the monoclonal antibodies anti-CD4 PerCP/Cy5.5 (BioLegend) and anti-CD8 Pacific Blue (BioLegend) for 30 min at 4 °C. After surface staining, cells were washed with FACS buffer, fixed and permeabilized using the Cytofix/Cytoperm kit (BD Biosciences). Cells were intracellularly stained with anti-CD3 APC-Cy7 (BD Biosciences), anti-TNF-α PE-Cy7 (BD Biosciences) anti-IFN-γ AF700 (BD Biosciences), and anti-IL-2 BV711 (BioLegend) mAbs for 30 min at 4 °C. Next, the cells were washed twice with the permeabilization buffer (BD Biosciences), fixed with 4% paraformaldehyde in PBS, and stored at 4 °C until analysis. Splenocytes were analyzed on a LSR II flow cytometer (BD Biosciences). 200,000-500,000 events were collected per specimen. Data were analyzed with the FlowJo 10 program. Data were expressed by subtracting the percentages of the unstimulated stained cells from the percentages of the peptide pool-stimulated stained samples.

## Vaccination and virus challenge
Female BALB/c mice (aged 6–8 weeks, The Jackson Laboratory for the Icahn School of Medicine at Mount Sinai and Charles River Laboratories for the University of Pennsylvania) were anesthetized and

shaved to expose the skin of the back. mRNA-LNP vaccines diluted to 5 µg per 100 µL (or 25 µg per 100 µL for the 5 µg pentavalent formulation) in PBS were injected intradermally (I.D.) into two sites distant from one another on the back to a total volume of 100 µL using a 31 G Ultra-Fine insulin syringe (BD). Mice vaccinated with TIV received 1 µg of the B/Malaysia/2506/2004 HA in 50 µL of PBS intramuscularly (I.M.) using a 28 G insulin syringe (BD). Four weeks after vaccination, mice were anesthetized and intranasally infected with 50 µL of influenza virus containing 1x or 5x the 50% mouse lethal dose (mLD$_{50}$). Additionally, mice were bled for serological analysis at this time point. Weight loss was monitored for 14 days after challenge, and mice that lost >25% of their initial body weight were humanely euthanized. For prime-boost studies mice were vaccinated twice as described above with an intervening interval of 4 weeks. Four weeks after the boost, mice were intranasally challenged with 5mLD$_{50}$ of B/Malaysia/2506/2004 (V).

For the duration of the experiment, mice were housed in individually ventilated cages on a 12 h dark/light cycle. The room was kept at 20 °C, 50% relative humidity. Food and water were provided ad libitum.

### Serum collection

Blood was collected via submandibular bleeding using 25-27 gauge needles (BD Biosciences) and 1.5 mL microtubes (Eppendorf) without anticoagulants. Blood was incubated at RT for 1 h and centrifuged at 2500 × g. Serum was separated from pellet and stored at 4 °C until used.

### Passive transfer of sera

Female BALB/c mice aged 6–8 weeks underwent a prime-boost regimen with 5 µg of mRNA vaccine per mouse with 4-week intervals between both vaccinations and subsequent whole blood harvest. Mice were anesthetized and then a cardiac puncture was performed to gather whole blood. The blood was allowed to coagulate at RT for 1 h before being placed at 4 °C for 30 min. Blood was then spun at 12,000 × g for 10 min at 4 °C, and sera were separated from remaining blood components and stored at 4 °C until further use. Naïve mice were intraperitoneally injected with 200 µL of sera 2 h prior to influenza virus challenge with 5mLD$_{50}$ of B/New York City/PV01181/2018 (V). Mice were bled post-transfer, and sera were tested against the appropriate antigen by enzyme-linked immunosorbent assay (ELISA) to ensure successful transfer.

### ELISA

Immulon plates (Immulon 4HBX; Thermo Scientific) were coated with 2 µg/mL recombinant protein (50 µL per well) in PBS at 4 °C O/N. The following day, the plates were washed 3 times with PBS containing 0.1% Tween-20 (PBS-T) and blocked in blocking solution (3% goat serum, 0.5% milk in PBS-T) for 1 h at RT. After blocking, pre-diluted serum was added to the first well to a final concentration of 1:100 in blocking solution. Serum was then serially diluted 1:3 in blocking solution and incubated at RT for 2 h. Plates were then washed three times with PBS-T before adding goat anti-mouse IgG HRP conjugate (Rockland) in blocking solution. The plates were incubated for 1 h at RT before being washed 4x with PBS-T with shaking. To develop the plates, 100 µL of O-phenylenediamine dihydrochloride (OPD) substrate (SigmaFast OPD; Sigma-Aldrich) was added to each well. After a 10-min incubation, the reaction was stopped by adding 50 µL of 3 M hydrochloric acid (HCl) to each well. The optical density at 490 nm (OD$_{490}$) was measured on a Synergy 4 plate reader (BioTek). A cut-off value of the average of the OD values of blank wells plus 3 standard deviations (SD) was established for each plate and used for calculating the area under the curve (AUC), which was the readout for this assay. The limit of detection of the assay was a titer of 1:100. Samples that did not reach this titer were assigned a value of 1:50.

For cell-based ELISAs, 96-well cell culture plates (Corning) were seeded with 100 µL/well of Madin-Darby canine kidney (MDCK) cells at a density of $2 \times 10^5$ cells/mL and incubated O/N at 37 °C with 5% CO$_2$. The following day, cells were washed with PBS and infected with 100 µL of B/Colorado/06/2017 at a multiplicity of infection (MOI) of 5 for 20 h at 33 °C with 5% CO$_2$. The cells were then fixed with 3.7% paraformaldehyde (Fisher Scientific) and stored at 4 °C for 24 h. The subsequent day, plates were washed, and cells were permeabilized with 0.1% Triton X-100 (Sigma-Aldrich) diluted in PBS and incubated for 15 min at RT; the cells were then washed twice with PBS and the remainder of the cell-based ELISA was performed as described above.

For M2 ELISAs, 96-well cell culture plates (Corning) were seeded with 100 µL/well of HEK293T cells at a density of $1 \times 10^5$ cells/mL and incubated O/N at 37 °C with 5% CO$_2$. The following day, cells were washed with PBS and transfected (TransIT-LT1, MirusBio) with 0.1 µg of plasmid DNA encoding B/Colorado/06/2017 M2 as per the manufacturer's instructions. The cells were then fixed with 3.7% paraformaldehyde in PBS (Fisher Scientific) and stored at 4 °C for 24 h. The subsequent day, plates were washed, and cells were permeabilized with 0.1% Triton X-100 (Sigma-Aldrich) diluted in PBS and incubated for 15 min at RT; the cells were then washed twice with PBS and the remainder of the cell-based ELISA was performed as described above.

### Microneutralization (MNT) assay

MDCK cells were seeded in 96-well cell culture plates (Corning) at a density of $2 \times 10^5$ cells/mL with a total volume of 100 µL in each well and incubated O/N at 37 °C in a humidified incubator with 5% CO$_2$. The following day, receptor destroying enzyme (RDE)-treated sera was serially diluted 2-fold across the plate in infection media (1x minimum essential media (MEM) (Gibco), 100 U/mL penicillin and 100 µg/mL streptomycin (Gibco), 10 mM 4-(2-hydroxyethyl)-1-piperazineethanesulfonic acid (HEPES) (Gibco), 2 mM L-glutamine (Gibco), 3.2% NaHCO$_3$ (Sigma-Aldrich), and 1.2% bovine serum albumin (BSA) (MP Biomedicals) supplemented with 1 µg/mL N-tosyl-L-phenylalanine chloromethyl ketone (TPCK) -treated trypsin (Sigma-Aldrich). Next, 60 µL of 100 × 50% tissue culture infectious dose (TCID$_{50}$) of virus in infection medium and 60 µL of serially diluted sera were incubated on a shaker for 1 h at RT. Before the end of the incubation time, MDCK cells were washed once with 200 µL PBS and then 100 µL of the incubated sera-virus mixture was added to the plates for 1 h at 33 °C with 5% CO$_2$. Afterwards, the virus inoculum was aspirated and MDCK cells were again washed with 200 µL PBS and incubated with 100 µL of the serially diluted sera for 72 h at 33 °C and 5% CO$_2$. Following the 3-day incubation, hemagglutination was assessed. In brief, 50 µL of cell supernatant was first added to V-bottom 96-well plates (ThermoFisher Scientific) with 50 µL of 0.5% chicken red blood cells (Lampire Biological Laboratories) diluted in PBS. The plates were incubated for 45 min at 4 °C. Data are displayed as the endpoint titer and this value represents the lowest dilution at which no hemagglutination could be detected.

### Antibody dependent cellular cytotoxicity (ADCC) reporter assay

ADCC reporter activity within the sera was tested by performing in vitro ADCC bioreporter assays as per the manufacturer's instructions (Promega). Briefly, $2 \times 10^5$ MDCK cells/mL were cultured O/N at 37 °C with 5% CO$_2$ in white, flat-bottom, 96-well cell culture plates (Corning). The following day the cells were washed with 200 µL PBS and infected with B/Colorado/06/2017 (V) or B/Phuket/3073/2013 (Y) at an MOI of 5. The plates were then incubated O/N at 33 °C with 5% CO$_2$. The next day sera were serially diluted 2-fold in RPMI 1640 (Gibco) and ADCC Bioassay effector cells were diluted to $3 \times 10^6$ cells/mL in RPMI 1640 media. Media was then aspirated from the cell plates and the cells were washed once with 100 µL/well PBS; 25 µL/well of assay buffer, 25 µL/well of serially diluted sera and 25 µL/well of ADCC effector cells (Promega)

were then added. ADCC effector cells are genetically engineered to express mouse FcγRIV receptor, the predominant receptor involved in ADCC in mice, and a luciferase reporter. Plates were incubated for 6 h at 33 °C with 5% $CO_2$ in a humidified incubator, followed by temperature equilibration for 15 min at RT and subsequent addition of 75 µL/ well Bio-Glo™ Luciferase assay reagent for 10 min in the dark at RT. After incubation, luminescence was assessed as a measure of luciferase expression using a Synergy H1 microplate reader (BioTek). A cut-off value of the average of the OD values of blank wells plus 5 SDs was established for each plate and used for calculating the AUC, which was the readout for this assay. The limit of detection of the assay was a titer of 1:10. Samples that did not reach this titer were assigned a value of 1:5 for graphing purposes.

### NA and NI assay

To determine NA activity, samples were tested on flat-bottom Immulon 4BX 96-well plates coated O/N at 4 °C with 100 µL of fetuin (Sigma) at 25 µg/mL in PBS. Fetuin-coated plates were then washed 3x with PBS-T. On a separate plate, viruses were serially diluted 2-fold in sample diluent (PBS (Gibco) with 0.9 mM $CaCl_2$ and 0.5 mM $MgCl_2$ supplemented with 1% bovine serum albumin (MP Biomedicals) and 0.5% Tween 20 (Fisher Scientific)); 100 µL of pre-diluted virus samples were added to the washed fetuin-coated plates. The fetuin-coated plates were then incubated for 18 h at 33 °C. Plates were then washed 3x with PBS-T and 100 µL/well of HRP-conjugated peanut agglutinin (PNA) in PBS were added to the plates. The plates were incubated for 2 h at RT before being washed 4x with PBS-T with shaking. To develop the plates, 100 µL of OPD substrate was added to each well. After a 10-min incubation, the reaction was stopped by adding 50 µL of 3 M HCl to each well. The $OD_{490}$ was measured on a Synergy 4 plate reader (BioTek). The half maximal effective concentration ($EC_{50}$) was determined using GraphPad Prism.

To measure NI, heat-treated sera (56 °C for 1 h) were diluted serially by 2-fold in sample diluent with a starting dilution of 1:100 and incubated for 18 h at 33 °C with an equal volume (50 µL) of the respective virus dilution in the fetuin-coated plates. The remainder of the assay was performed as described above. One column on the plate contained sample diluent without antibody and served as a positive (virus-only) control. Another column contained sample diluent only (no virus) and served as a negative (background) control. Data were analyzed in GraphPad Prism 8.

### Plaque assays

Virus titers were determined by plaque assay on MDCK cell monolayers in 12-well plates (Corning). Allantoic fluid and lung homogenates were serially diluted by 10-fold in PBS and incubated on MDCK cells for 1 h before the addition of an agarose overlay containing a final concentration of 0.64% agarose (Oxoid), 1 x minimum essential medium (MEM) (10% 10xMEM (Gibco), 2 mM L-glutamine, 0.1% of sodium bicarbonate (wt/vol; Gibco), 10 mM HEPES, 100 U/ml penicillin–100 µg/ml streptomycin, 0.2% BSA, 1 µg/mL TPCK-treated trypsin, and 0.1% (wt/vol) DEAE (diethylaminoethyl)-dextran was added to the cells. The cells were then incubated for 72 h at 33 °C, and visible plaques were counted after fixation with 3.7% formaldehyde in PBS and visualization with a crystal violet counterstain (Sigma-Aldrich). All virus titers are presented as the $\log_{10}$ plaque forming units (PFU)/mL. The limit of detection for these assays was 50 PFU/ mL. The AUC values for virus titers over the time course of infection were calculated using the $\log_{10}$ PFU/mL, and the AUC value was calculated using GraphPad Prism 8 software.

### Determination of amino acid similarity

The HA, NA, NP and M2 sequences from B/Colorado/06/2017 (V) (EPI_ISL_277231), B/Phuket/3073/2013 (Y) (EPI_ISL_517766), B/Lee/1940 (NIGSP_BUCB_00001), B/Malaysia/2506/2004 (V) (EPI_ISL_29398), B/

Florida/04/2006 (Y) (EPI_ISL_142731), B/New York City/PV01181/2018 (V), and B/New York City/PV00094/2017 (Y) (OFL_ISL_214386) were assessed using the Clustal Omega multiple sequence alignment tool[51].

### Statistics and reproducibility

For line graphs data are expressed as means. For bar graphs data are expressed as individual values and the average presented as the mean. Error are represented by SD or standard error of the mean (SEM). Significance between antibody responses were analyzed by one-way ANOVA followed by a multiple comparison test. A two-tailed unpaired t-test was applied to determine statistical significance in T cell assays. Groups were compared to the luciferase control group. Data were considered statistically significant at $p < 0.05$. All statistical analyses were performed using GraphPad Prism 8.

Experiments were conducted with sufficient power ($n = 5$ mice/ group) to ensure distinct differences between the groups to be detected.

### Reporting summary

Further information on research design is available in the Nature Research Reporting Summary linked to this article.

## Data availability

Source data are available with this paper. Source data are provided with this paper.

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

## Acknowledgements

The study was partially supported by NIH R01-AI146101 (N.P. and M.M.) and by the Collaborative Influenza Vaccine Innovation Centers (CIVICs)

contract 75N93019C00051 (F.K.). We thank the Mount Sinai Pathogen Surveillance Program, and especially Viviana Simon and Harm van Bakel, for sharing influenza virus isolates B/New York City/PV00094/2017 and B/New York City/PV01181/2018. The overlapping peptide pools used in the T cell studies were kindly provided by Michael J. Hogan and Laurence C. Eisenlohr (CHOP). Figure 5a was created with BioRender.

## Author contributions

N.P. and M.M. conceptualized the study. H.M. produced mRNA vaccine antigens. Cs.B. performed cell transfections, Western blotting, the T cell assays and flow cytometry. F.K. provided oversight on studies performed at the Icahn School of Medicine at Mount Sinai. Y.K.T. and M.M.H.S. encapsulated mRNAs into LNPs. J.M.C., J.T., G.O., W.R., S.S., D.B., M.L. and M.M. performed animal experiments and serological assessment. M.M. and N.P. wrote the paper with help from co-authors.

## Competing interests

In accordance with the University of Pennsylvania policies and procedures and our ethical obligations as researchers, we report that NP and YKT are named on a patent describing the use of nucleoside-modified mRNA in lipid nanoparticles as a vaccine platform. NP and FK are named on a patent filed on universal influenza vaccines using nucleoside-modified mRNA. FK is also named on several patents and patent applications for universal influenza virus vaccine candidates based on other vaccine platforms. We have disclosed those interests fully to the University of Pennsylvania and The Icahn School of Medicine at Mount Sinai, and we have in place an approved plan for managing any potential conflicts arising from licensing of our patents. MMHS and YKT are employees of Acuitas Therapeutics, a company focused on the development of LNP nucleic acid delivery systems for therapeutic applications. FK has consulted for Merck and Pfizer (before 2020) and currently consults for Pfizer, Seqirus, and Avimex. MM is now employed at Seqirus, Parkville, Australia. The remaining authors declare no competing interests.

## Additional information

**Peer review information** *Nature Communications* thanks Yasushi Itoh and other anonymous reviewer(s) to the peer review of this work. Peer review reports are available.

