## [Peer Review File · Nature Communications]

REVIEWER COMMENTS

Reviewer #1 (Remarks to the Author):

The manuscript Pardi et al., details the use of nucleoside modified mRNA vaccines where the authors compare different formulations to understand the cross protection of B influenza viruses. The results presented demonstrate the applicability of such an approach with naive mice indicating the importance of the development of sufficient antibodies to elicit significant cross protection. In this respect the identification that antibodies are essential for cross protection is not novel since this is well established in the field. The authors however, have utilised mRNA LNPs in order to elicit such a response and demonstrated findings. The methodology, conclusions, data analysis is sound and the details as to the methodology is sufficient for the field to repeat or extend this work further.

The work presented is limited, however, to naive mice and the authors should modified the discussion to reflect this since it would be unexpected that humans that have had prior exposure may have identical responses and it is unclear whether prior exposure would lead to altered more biased responses that do not display the cross reactivity that has been demonstrated in a naive animal population.

I think the manuscript is acceptable with minor revisions as outlined.

Minor revisions that would enhance this manuscript:

The statement “These vaccines elicit an antibody response towards the HA.” is correct yet but many QIV vaccines contain B NA and there are demonstrable antibodies to B NA. The sentence should be altered to “vaccines are focused to elicit an antibody response” (ref 5,8,9 of this article).

That lineage-matched HA were also able to exhibit NAI due to steric hindrance mediated by HA stem antibodies. While this has been referenced, the observation may also be due to anti-carbohydrate specific antibodies. Was this investigated? This should be revised in the manuscript if not.

Reviewer #2 (Remarks to the Author):

The study by Pardi and colleagues addresses the potency of a multivalent mRNA vaccine for Influenza B virus strains. They present data from various mRNA constructs on their generation of neutralizing antibody titers and in vivo protection after challenge in mice. They also perform serum transfer experiments resulting in protection after challenge which is important data demonstrating that antibodies are critical after mRNA vaccination to influenza B. Functional studies of this kind are timely and have large implications in the development of mRNA vaccines against influenza. The methodology is sound and the data presented clearly.

Comments:

1. The major short-coming is the lack of the respective protein vaccine control groups or benchmark licensed vaccines, at least in some form. Such data would make a stronger demonstration of how a more conventional vaccine platform performs against the mRNA platform. Also this would be an important comparison regarding prime-boost regimens and whether titers after prime are different. The mRNA platform has clear advantages over the protein vaccine in production etc but it would be much valued information on differences in kinetics and quality of the immune responses.

2. Fig 8. Single immunization using dose range is a good informative experiment. The authors conclude that a low dose protects in mice. However, what is the evidence what should be considered a low dose? Can the data/doses be translated to humans in any way?

3. The expression one-shot vs double-shot appears as slang and should be considered to be replaced e.g. one-dose or similar.

Reviewer #3 (Remarks to the Author):

In the present manuscript, the efficacy of mRNA vaccine against influenza B virus was examined. Antibody responses were assessed by MNT, NAI, and ADCC using Yamagata and Victoria lineage viruses. mRNA-LNP induced antibody responses effectively. However, the authors did not examine antigen-specific responses of helper T cells and CTL.

1. At immunization, what kind of needle was used for intradermal injection?
2. Describe ADCC Bioassay effector cells in detail.

3. Did the pentavalent vaccine induce formation of virus-like particles or microsomes containing a set of viral proteins? If so, the possibility that particle-form antigens are more immunogenic should be discussed.
4. Fig. 4 and Fig. 5 should be presented in one figure.
5. In general, pentavalent vaccination induced higher responses in MNT, NAI, and ADCC and protection than did monovalent HA and NA vaccinations. The reason should be discussed.
6. How did the vaccine of NP mRNA work for protection? Furthermore, how did antibody against NP work, although NP is supposed to present inside the cells not on the surface of the cells? The discussion is required in addition to lines 521-535.
7. The analysis of antigen-specific responses of helper T cells and CTLs should be added.

Response to Reviews

Thank you for considering our manuscript (NCOMMS-21-47367) entitled “Development of a pentavalent broadly protective nucleoside-modified mRNA vaccine against influenza B viruses” for publication in Nature Communications. We appreciate the critical reading by the editor and the reviewers, and we believe that all concerns have been appropriately addressed. Please find our responses to the comments in red below.

Reviewer #1 (Remarks to the Author):

The manuscript Pardi et al., details the use of nucleoside modified mRNA vaccines where the authors compare different formulations to understand the cross protection of B influenza viruses. The results presented demonstrate the applicability of such an approach with naive mice indicating the importance of the development of sufficient antibodies to elicit significant cross protection. In this respect the identification that antibodies are essential for cross protection is not novel since this is well established in the field. The authors however, have utilised mRNA LNPs in order to elicit such a response and demonstrated findings. The methodology, conclusions, data analysis is sound and the details as to the methodology is sufficient for the field to repeat or extend this work further.

1. The work presented is limited, however, to naive mice and the authors should modified the discussion to reflect this since it would be unexpected that humans that have had prior exposure may have identical responses and it is unclear whether prior exposure would lead to altered more biased responses that do not display the cross reactivity that has been demonstrated in a naive animal population.

We agree with the reviewer and have modified the discussion. Please see lines 342-346 in the revised manuscript.

I think the manuscript is acceptable with minor revisions as outlined.

Minor revisions that would enhance this manuscript:

2. The statement “These vaccines elicit an antibody response towards the HA.” is correct yet but many QIV vaccines contain B NA and there are demonstrable antibodies to B NA. The sentence should be altered to “vaccines are focused to elicit an antibody response” (ref 5,8,9 of this article).

We agree, “These vaccines elicit an antibody response towards the HA.” was modified to “These vaccines are focused on eliciting an antibody response towards the HA.” Please see line 52 in the modified document.

3. That lineage-matched HA were also able to exhibit NAI due to steric hindrance mediated by HA stem antibodies. While this has been referenced, the observation may also be due to anti-carbohydrate specific antibodies. Was this investigated? This should be revised in the manuscript if not.

Anti-carbohydrate specific antibodies were not investigated. The manuscript was revised to include “This could be due to steric hindrance mediated by anti-HA antibodies (17) or by anti-carbohydrate antibodies (18).” Please see lines 140-141 in the modified manuscript.

Reviewer #2 (Remarks to the Author):

The study by Pardi and colleagues addresses the potency of a multivalent mRNA vaccine for Influenza B virus strains. They present data from various mRNA constructs on their generation of neutralizing antibody titers and in vivo protection after challenge in mice. They also perform serum transfer experiments resulting in protection after challenge which is important data demonstrating that antibodies are critical after mRNA vaccination to influenza B. Functional studies of this kind are timely and have large implications in the development of mRNA vaccines against influenza. The methodology is sound and the data presented clearly.

Comments:

1. The major short-coming is the lack of the respective protein vaccine control groups or benchmark licensed vaccines, at least in some form. Such data would make a stronger demonstration of how a more conventional vaccine platform performs against the mRNA platform. Also this would be an important comparison regarding prime-boost regimens and whether titers after prime are different. The mRNA platform has clear advantages over the protein vaccine in production etc but it would be much valued information on differences in kinetics and quality of the immune responses.

We completely agree with the reviewer and performed a challenge experiment with B/Malaysia/2506/2004 (V) following a prime-only vaccination with monovalent mRNA-LNPs (5 µg) or with 5 µg/antigen or 1 µg/antigen of the pentavalent mRNA-LNP formulation. We included control animals receiving 5 µg of luciferase mRNA-LNP or trivalent influenza virus vaccine (TIV, Fluzone, 2006-2007) at 1 µg of matched HA. Our results indicate that in all cases the pentavalent mRNA-LNPs, either at 5 µg or at the lower dose, outperform the TIV (please see Supplementary Fig. 2 in the revised paper). High antibody titers were detected against B/Phuket/3073/2013 HA (Y), B/Colorado/06/2017 HA (V), or B/Colorado/06/2017 NA (V) following administration of the pentavalent formulations, while TIV only induced detectable antibodies against B/Colorado/06/2017 HA (V). The TIV formulation used here contains the B/Malaysia/2506/2004 strain, hence we were interested in performing a challenge with this virus. In line with our previous observations (Fig. 6 in the revised manuscript), luciferase controls and M2-immunized mice displayed severe morbidity and succumbed to infection, while mice in all the other groups survived. Notably, TIV-vaccinated mice, although survived the challenge, displayed marked weight loss that was not completely recovered by the end of the observation period. Our new results are described starting at line 193 in the revised manuscript.

2. Fig 8. Single immunization using dose range is a good informative experiment. The authors conclude that a low dose protects in mice. However, what is the evidence what should be considered a low dose? Can the data/doses be translated to humans in any way?

We agree and more discussion has been added to the revised paper. Please see lines 355-359 in the revised manuscript.

3. The expression one-shot vs double-shot appears as slang and should be considered to be replaced e.g. one-dose or similar.

We agree with the reviewer and one-shot and double-shot were replaced with prime-only and prime-boost, respectively, throughout the manuscript and in figures.

Reviewer #3 (Remarks to the Author):

In the present manuscript, the efficacy of mRNA vaccine against influenza B virus was examined. Antibody responses were assessed by MNT, NAI, and ADCC using Yamagata and Victoria lineage viruses. mRNA-LNP induced antibody responses effectively. However, the authors did not examine antigen-specific responses of helper T cells and CTL.

1. At immunization, what kind of needle was used for intradermal injection?

BD Ultra-Fine insulin syringes (31G) were used for intradermal injections. This information was added at line 557.

2. Describe ADCC Bioassay effector cells in detail.

A description of the cells was added at lines 648-649.

3. Did the pentavalent vaccine induce formation of virus-like particles or microsomes containing a set of viral proteins? If so, the possibility that particle-form antigens are more immunogenic should be discussed.

We thank the reviewer for this suggestion. As reported by other groups, the HA, NA, matrix protein 1 (M1), and M2 are important for virus like particles (VLPs) formation (Wu et al PMID: 20339535, Latham et al PMID: 11390617). Particularly the M1, is critical for VLP formation, and it seems to be the only protein that can form VLP-like structures by itself (Latham et al PMID: 11390617). Although we have not tested experimentally if our pentavalent formulations are able to form VLPs *in vitro* or *in vivo*, given that M1 is not included in the mixture, it is unlikely that these types of structures are formed.

4. Fig. 4 and Fig. 5 should be presented in one figure.

We agree with the reviewer, Fig. 4 and Fig. 5 have been included as one figure (Fig. 4 in the revised manuscript). Figure references and captions were edited throughout the manuscript.

5. In general, pentavalent vaccination induced higher responses in MNT, NAI, and ADCC and protection than did monovalent HA and NA vaccinations. The reason should be discussed.

We thank the reviewer for this suggestion. We have modified the text: "The modest improvement we observed in the pentavalent formulations may be attributed to the combined multiantigen targeted immune response as described below for each antigen." Please see lines 366-368 in the revised manuscript.

6. How did the vaccine of NP mRNA work for protection? Furthermore, how did antibody against NP work, although NP is supposed to present inside the cells not on the surface of the cells? The discussion is required in addition to lines 521-535.

We agree with the reviewer and discussion regarded this was added at lines 416-431 in the revised manuscript.

7. The analysis of antigen-specific responses of helper T cells and CTLs should be added.

We agree with the reviewer and we have measured antigen-specific T cell responses against B/Phu HA, B/Col HA, B/Col NA and B/Col NP. We found that all vaccines induced antigen-specific CD4⁺ and CD8⁺ T cell responses after administration of a single vaccine dose. Please see Fig. 5 and Supplementary Fig. 1 and lines 160-167 in the revised manuscript.

REVIEWERS' COMMENTS

Reviewer #2 (Remarks to the Author):

My comments were appropriately addressed.

Reviewer #3 (Remarks to the Author):

The authors added the results for T-lymphocyte responses specific for virus peptides. mRNA-LNP vaccination induced CD4+ and CD8+ T lymphocytes specific for antigen.

The authors also added methods on flow cytometer, a vaccination needle, and ADCC.